# Phase-Preserving Analytical Features from Solid Harmonic Wavelet Bispectrum Simplify Decision Boundaries

## Abstract

We introduce the Solid Harmonic Wavelet Bispectrum, an operator for 2D images that computes third-order correlations over angular frequency components of solid harmonic wavelet responses. By using angular rather than spatial frequencies, our bispectrum achieves lower dimensionality than traditional 2D scattering-based bispectra, avoiding comparisons across two spatial dimensions while still preserving rich frequency information. Extending these bispectra to first- and second-order scattering coefficients produces low-dimensional multi-scale features that capture detailed image structure. To illustrate the quality of the representations, we use k-nearest neighbors, which highlights that our features encode meaningful similarity structure even without a learned parametric classifier. Results on texture, medical, and galaxy images demonstrate that these features show improved separability and similarity structure compared to existing geometric and deep learning-based representations.

## 1 Introduction

Designing discriminative features that remain robust to geometric variability is essential in supervised learning. A common strategy is to exploit symmetries—transformations such as translations and rotations that preserve labels—by constructing representations that are covariant under these actions and then pooling to obtain invariants. This principle underlies many methods in harmonic analysis and group-based learning, and is particularly valuable in domains like computational chemistry, astronomy, and medical imaging, where data is costly and interpretability matters.

Convolutional Neural Networks (CNNs) achieve strong performance through end-to-end optimization, but they lack built-in geometric priors and typically require large datasets and augmentation. Geometric deep learning introduces equivariant architectures to encode symmetry, yet these approaches depend on predefined groups, large parametric models, and heavy training, which can reduce transparency and compromise analytical guarantees. By contrast, wavelet scattering networks (Bruna & Mallat, 2013) provide invariance by design using cascades of predefined filters, modulus nonlinearities, and averaging. Wavelet Scattering operators have matched deep learning baselines with simple classifiers in tasks such as texture discrimination (Sifre & Mallat, 2014), quantum molecular regression (Eickenberg et al., 2017), and object classification (Oyallon et al., 2014). A key limitation, however, is that the modulus operator discards phase information, restricting scattering networks to second-order statistics.

We introduce the *Solid Harmonic Wavelet Bispectrum* (SHWB), an operator that unifies multi-scale analysis, higher-order statistics, and group invariance. Bispectral analysis preserves phase couplings lost by power spectra or modulus operators, but prior formulations in the Fourier domain were not adapted to geometric invariance. Our approach computes third-order correlations over solid harmonic wavelet responses, capturing nonlinear interactions between rotational components at each scale. By spanning one angular frequency instead of two spatial frequencies, SHWB yields compact, frequency-preserving representations that are covariant to roto-translations and can be integrated to form invariants. Extending bispectral analysis to first- and second-order scattering coefficients further enriches the representation while keeping dimensionality low.

A key methodological choice is to evaluate SHWB features in isolation with simple classifiers, following prior work on scattering (Bruna & Mallat, 2013; Eickenberg et al., 2017) and linear probes for representation learning (Alain & Bengio, 2016). This avoids conflating the quality of the representation with classifier capacity, reduces computational cost, and highlights the intrinsic discriminative power of the features. Despite this minimalist evaluation, SHWB achieves competitive or superior performance to deep learning baselines across textures, medical images, and a $\sim$20K-image astrophysics dataset, where simple models such as k-nearest neighbors already reveal strong similarity structure in the proposed feature space.

## 2 RELATED WORK

**Wavelet and Scattering Methods.** Invariant feature extraction has long been studied in harmonic analysis. Multiresolution analysis (Mallat, 1989) and steerable wavelets (Freeman & Adelson, 1990; Simoncelli et al., 1992) decompose signals into multi-scale orientation-separable filters, providing translation and rotation invariance. Wavelet scattering networks (Bruna & Mallat, 2013) extend this with cascades of wavelet transforms, modulus nonlinearities, and spatial averaging, offering stability to deformations while retaining interpretability. Steerable (Sifre & Mallat, 2013) and solid harmonic wavelets (Eickenberg et al., 2017) further capture roto-translation invariants, achieving strong results in texture classification, molecular regression, and related tasks. The main drawback is that the modulus operator discards phase and restricts scattering to second-order statistics, although Mallat et al. (2019) showed how phase correlations can be reintroduced in sparse signals. Recent work on scattering spectra Cheng et al. (2024) extends scattering to higher-order statistics for generative modeling of physical systems, using maximum entropy models and covariances of scattering coefficients to generate new field realizations. Our work extends this line by incorporating bispectral statistics into solid harmonic scattering to preserve discriminative phase interactions for supervised learning, computing bispectral coefficients explicitly while maintaining analytical rotation invariance and interpretability. Other recent developments include Rodriguez et al. (2019), who augmented scattering with learned CNN features for image classification, and Saydjari & Finkbeiner (2022), who developed efficient computational algorithms for scattering transforms. Our approach differs by focusing on analytical bispectral features without learned parameters.

**Geometric Deep Learning.** Geometric deep learning develops neural architectures that respect the symmetries of data by ensuring equivariance to group actions. Group-equivariant CNNs (G-CNNs) (Cohen & Welling, 2016) and harmonic networks (H-Nets) (Worrall et al., 2017) exemplify this approach, where feature maps transform predictably under rotations or translations. Invariance is typically obtained only after additional pooling or integration over the group. Bispectral neural networks (Sanborn et al., 2023) move closer to invariance by learning bispectral coefficients that collapse transformed inputs into identical representations, but this requires large parametric models and orbit-based supervision. By contrast, our method produces invariants analytically and by construction, without training or group-specific pooling, allowing simple classifiers to operate directly on the features while retaining interpretability and computational efficiency.

**Higher-Order Spectra.** Higher-order statistics, such as the bispectrum, capture phase correlations lost by second-order statistics. Higher-order spectral analysis originates from studies of non-Gaussian signals, with applications in fields such as oceanography, telecommunications, and plasma physics (Nikias & Raghuveer, 1987). This was extended to multi-scale bispectral analysis over wavelet transforms (van Milligen et al., 1995; Jamšek et al., 2007; Jamšek et al., 2010), while Newman et al. (2021) proposed a normalised "bispectral density" subject to wavelet parametrisation. However, such approaches remain sensitive to rotation, lacking geometric invariance. On the other hand, Kakarala (2012) projected signals onto solid harmonics before computing the Fourier bispectrum. Kondor (2007) projected images onto the unit sphere and decomposed them into solid harmonics, though at the cost of distortions due to spherical projection and redundant computation.

While prior work has addressed invariance or higher-order statistics separately, few methods unify both in an interpretable, efficient, and empirically competitive manner. SHWB aims to fill this gap.

## 3 SOLID HARMONIC WAVELET BISPECTRUM

### 3.1 BACKGROUND: SOLID HARMONIC WAVELET TRANSFORM

The Solid Harmonic Wavelet Transform (SHWT) Eickenberg et al. (2017) provides multiscale roto-translation covariant features for 2D and 3D signals. Solid harmonics, solutions of the Laplace equation in polar or spherical coordinates, form a complete basis on the unit circle or sphere. Localized solid harmonic wavelets are obtained by modulating them with a Gaussian envelope, yielding band-pass, zero-mean filters.

In 2D, a wavelet at scale $j$ and angular frequency $\ell$ is

$$\psi_{j,\ell}(r,\varphi) = \frac{1}{\sqrt{(2\pi\sigma_j^2)^2}} e^{-r^2/2\sigma_j^2} r^\ell e^{i\ell\varphi}, \qquad \sigma_j = 2^j \sigma_0, \tag{1}$$

normalized to unit energy. Varying $j$ and $\ell$ produces a filter bank; the case $\ell = 0$ reduces to an isotropic Gaussian low-pass. This aligns with the principles of multiresolution analysis (Mallat, 1989), enabling the capture of both coarse- and fine-scale structures. A visualization of the resulting wavelets is included in Appendix A.2.

For a signal $x$ and location $u = (r\cos\varphi, r\sin\varphi)^T$, the solid harmonic wavelet transform at scale $j$ and frequency $\ell$ is

$$W[x,j,\ell](u) = (x \star \psi_{j,\ell})(u). \tag{2}$$

The Solid Harmonic Wavelet Scattering transform applies a modulus non-linearity to remove phase and yield roto-translation covariance,

$$U[x,j,\ell](u) = |x \star \psi_{j,\ell}|(u). \tag{3}$$

Higher-order features are obtained by cascading this operation, e.g.

$$U[x,j_1,\ell_1,j_2,\ell_2](u) = \big|U[x,j_1,\ell_1] \star \psi_{j_2,\ell_2}\big|(u). \tag{4}$$

Integration over $u$ reduces these covariant responses to invariants spanning scales $j_*$ and angular frequencies $\ell_*$. However, the modulus discards phase information, which motivates redundant filter-bank designs and dedicated phase-recovery methods Waldspurger et al. (2015). In contrast, higher-order spectra explicitly reintroduce phase relationships by encoding nonlinear interactions between frequency components. We review these next.

### 3.2 BACKGROUND: HIGHER-ORDER SPECTRA

Second-order statistics, such as the power spectrum $P(f) = |\hat{x}(f)|^2$, describe the distribution of energy across frequencies but discard phase information, limiting their ability to represent structure in natural signals (Oppenheim & Lim, 1981; Skarbnik et al., 2010).

Higher-order spectra extend to third- and fourth-order statistics, namely the bispectrum and trispectrum. The bispectrum captures quadratic phase coupling among components $f_1$, $f_2$, and $f_1 + f_2$:

$$B(f_1, f_2) = \hat{x}(f_1)\,\hat{x}(f_2)\,\overline{\hat{x}(f_1 + f_2)}. \tag{5}$$

If we write the above terms as complex numbers of amplitude $a_f$ and phase $\varphi_f$, we have $a_{f_1} a_{f_2} a_{f_1+f_2} e^{i(\varphi_{f_1} + \varphi_{f_2} - \varphi_{f_1+f_2})}$. Therefore, the bispectrum maintains information about the phase interactions but its magnitude also reflects underlying power at each frequency. To separate coupling from power, one defines the *bicoherence*:

$$\text{Bic}(f_1, f_2) = \frac{|B(f_1, f_2)|}{\sqrt{|\hat{x}(f_1)|^2 |\hat{x}(f_2)|^2 \, |\hat{x}(f_1 + f_2)|^2}}, \tag{6}$$

which is bounded between 0 and 1, with values near 1 indicating strong phase coupling.

### 3.3 Solid Harmonic Wavelet Bispectrum

The Solid Harmonic Wavelet Bispectrum (SHWB) extends the scattering framework by capturing higher-order, phase-preserving interactions between solid harmonic wavelet responses across angular frequencies. For a signal $x$, at scale $j$ and angular frequencies $\ell_1, \ell_2$, we define the bispectrum in the Fourier domain as

$$\text{SHWB}[x, j, \ell_1, \ell_2](f) = \hat{W}[x, j, \ell_1](f) \cdot \hat{W}[x, j, \ell_2](f) \cdot \overline{\hat{W}[x, j, \ell_1 + \ell_2](f)}, \quad (7)$$

where $f$ indexes spatial frequencies (horizontal and vertical), and $\hat{W}[x, j, \ell]$ is the Fourier transform of the wavelet response.

As shown by Eickenberg et al. (2017), a rotation $R_\gamma \in \text{SO}(2)$ of angle $\gamma$ applied to a signal $x$ acts on the solid harmonic wavelet responses by a multiplicative phase factor:

$$W[R_\gamma x, j, \ell] = e^{i\ell\gamma} R_\gamma W[x, j, \ell]. \quad (8)$$

Substituting Equation 8 into 7, and using the constraint $\ell_3 = \ell_1 + \ell_2$, the phase factors cancel, yielding rotational covariance:

$$\text{SHWB}[R_\gamma x, j, \ell_1, \ell_2](f) = R_\gamma \text{SHWB}[x, j, \ell_1, \ell_2](f). \quad (9)$$

so the bispectrum is rotation-covariant. In contrast to scattering networks with Morlet wavelets, which achieve invariance by discarding phase via a modulus nonlinearity, SHWB **retains complex phase** information while still achieving covariance.

**Roto-translation invariance and $L_p$ integration.** Integration over spatial frequencies produces roto-translation invariance: while rotation of the input $x$ rotates the representation in the frequency domain, the integral over all frequencies remains unchanged. For real-valued signals, Hermitian symmetry $(\hat{x}(-f) = \overline{\hat{x}(f)})$ ensures that this integral yields a real-valued scalar:

$$\text{SHWB}[x, j, \ell_1, \ell_2] = \int_f \text{SHWB}[x, j, \ell_1, \ell_2](f)\, df. \quad (10)$$

To enrich the representation and capture different aspects of the spatial-frequency distribution, we compute these integrals using various $L_p$ norms:

$$\text{SHWB}[x, j, \ell_1, \ell_2, p] = \Big( \sum_f \big| \text{SHWB}[x, j, \ell_1, \ell_2](f) \big|^p \Big)^{\frac{1}{p}}, \quad (11)$$

Varying $p$ provides estimates of different moments of the bispectral distribution, enhancing sensitivity to the structure of nonlinear feature interactions.

**Bicoherence.** Normalizing by the magnitude of each term isolates the strength of phase alignment independently of signal amplitude:

$$\text{SHWBic}[x, j, \ell_1, \ell_2] = \frac{\big| \text{SHWB}[x, j, \ell_1, \ell_2] \big|}{\sqrt{\int |\hat{W}[x, j, \ell_1](f)|^2 df \ \int |\hat{W}[x, j, \ell_2](f)|^2 df \ \int |\hat{W}[x, j, \ell_1 + \ell_2](f)|^2 df}} \quad (12)$$

which takes values in $[0, 1]$ with 1 indicating strong phase coupling between $\ell_1$ and $\ell_2$.

#### 3.3.1 Comments on Computational Efficiency

A critical observation is that by computing the bispectrum over angular frequencies rather than Cartesian frequency pairs, SHWB efficiently spans the 2D space of nonlinear interactions with coefficients that scale quadratically with the number of angular frequencies as opposed to quartic scaling in the spatial dimensions.

Additionally, to reduce computational cost, we apply the following strategies:

- **Precomputation:** Wavelet responses $W[x, j, \ell]$ are computed once and cached, reused across multiple bispectral terms. Mini-batching manages memory for large datasets.
- **Symmetry:** SHWB is symmetric in $\ell_1$ and $\ell_2$, so we compute only $\ell_1 < \ell_2$.
- **Acceleration:** Computations are implemented in PyTorch with FFT-based convolutions, leveraging GPU acceleration and the Kymatio framework Andreux et al. (2018).

### 3.4 Solid Harmonic Wavelet Scattering Bispectrum

The scattering transform, and solid harmonic wavelet scattering by extension, attempt to obtain high order information by a cascade of wavelets in a manner similar to Deep Learning. In this section, we demonstrate that this approach, while proposed as an alternative to higher order spectra, can easily be combined with higher order spectra to provide rich feature spaces. We do that by demonstrating that the bispectrum can be used as an alternative non-linearity at the last layer/order of a scattering transform, while maintaining a cascade of wavelet convolutions and modulus non-linearities for intermediate layers.

As in relevant literature (Mallat, 2012; Eickenberg et al., 2017), we compute the bispectrum over first- and second- order scattering coefficients. The zeroth-order bispectral coefficient is computed as in other scattering transforms using the signal integral, or in practice by calculating $L_p$ to capture different proxies for the raw moments of the distribution of the data:

$$\text{SHWSB}_0[x, p] = \|x\|_p \tag{13}$$

For the first and second order coefficients, we introduce the following notation for the linear responses of the first- and second-order scattering:

$$W[x, j_1, \ell](u) = x \star \psi_{j_1, \ell}(u) \tag{14}$$

$$W[x, j_1, j_2, \ell](u) = |x \star \psi_{j_1, \ell}| \star \psi_{j_2, \ell}(u) \tag{15}$$

For all $\ell_1, \ell_2 \in L$ with $\ell_1 < \ell_2$, and $\ell_1 + \ell_2 = \ell_3 \in L$, we define the bispectral coefficients:

$$\text{SHWSB}[x, j_1, \ell_1, \ell_2](f) = \hat{W}[x, j_1, \ell_1](f) \cdot \hat{W}[x, j_1, \ell_2](f) \cdot \overline{\hat{W}[x, j_1, \ell_3]}(f) \tag{16}$$

$$\text{SHWSB}[x, j_1, j_2, \ell_1, \ell_2](f) = \hat{W}[x, j_1, j_2, \ell_1](f) \cdot \hat{W}[x, j_1, j_2, \ell_2](f) \cdot \overline{\hat{W}[x, j_1, j_2, \ell_3]}(f), \tag{17}$$

for first and second order coefficients. The first order invariant, $\text{SHWSB}[x, j_1, \ell_1, \ell_2]$, and second order invariant, $\text{SHWSB}[x, j_1, j_2, \ell_1, \ell_2]$, are obtained through integration over $f$ in Equations 16 and 17 respectively. Before integration, the coefficients remain complex-valued, encoding relative phase information across angular frequencies; after integration, Hermitian symmetry ensures that, for real signals, the resulting invariants are real-valued. The bicoherence (SHWSBic) is again derived by normalising with the product of the scattering magnitudes:

$$\text{SHWSBic}[x, j_1, \ell_1, \ell_2] = \frac{\text{SHWSB}[x, j_1, \ell_1, \ell_2]}{\|\hat{W}[x, j_1, \ell_1]\| \cdot \|\hat{W}[x, j_1, \ell_2]\| \cdot \|\hat{W}[x, j_1, \ell_3]\|} \tag{18}$$

$$\text{SHWSBic}[x, j_1, j_2, \ell_1, \ell_2] = \frac{\text{SHWSB}[x, j_1, j_2, \ell_1, \ell_2]}{\|\hat{W}[x, j_1, j_2, \ell_1]\| \cdot \|\hat{W}[x, j_1, j_2, \ell_2]\| \cdot \|\overline{\hat{W}[x, j_1, j_2, \ell_3]}\|} \tag{19}$$

As with SHWB, the SHWSB transform can perform the integration over spatial frequencies for Equations 16–17 by applying $L_p$ pooling on the spatial coordinates of the representations, which we denote with the parameter $p$ as:

$$\text{SHWSB}[x, j_1, \ell_1, \ell_2, p] = \left( \sum_f \left| \text{SHWSB}[x, j_1, \ell_1, \ell_2](f) \right|^p \right)^{\frac{1}{p}} \tag{20}$$

When $L_p$ pooling is applied, the normalization in Equations 18 and 19 is adjusted by replacing raw magnitudes with their pooled values, ensuring that SHWSBic remains bounded in $[0, 1]$.

First-order bispectral coefficients are equivalent to our definition of SHWB, and so the property in Equation 9 holds - they are roto-translation covariant by definition, and reduce to invariants through $L_p$ pooling. The formal proof of roto-translation covariance is detailed in Appendix A.3.

Second-order scattering coefficients retain the modulus nonlinearity after the first convolution, preserving stability. Although the modulus discards inter-scale phase, we argue that this phase loss is offset by increased robustness, consistent with scattering theory. Phase interactions within the same scale are retained through the bispectral term. Under rotation, a signal's convolution with a solid harmonic wavelet transforms by a phase factor, as shown in Equation 8. The modulus operator applied on the right side of Equation 8 removes this factor, leaving:

$$U[R_\gamma x, j, \ell](u) = R_\gamma(x \star \psi_{j, \ell})(u) \tag{21}$$

Since solid harmonics commute with rotation, a second convolution preserves rotation structure. Therefore, the same property as in Equation 9 establishes that both first- and second-order SHWSB coefficients are rotation covariant. $L_p$ pooling converts them to invariants.

## 4 EXPERIMENTS

We benchmark using simple models, parametrising $\ell$ over a coarse sweep and selecting $j$ relative to the scale of features relevant to the task. In each experiment, we report the best result for each representation, avoiding arbitrary parameter choices that best suit our new operators. The main strategy of the experiments is to demonstrate the ability of the Solid Harmonic Transforms to identify novel and interesting image features. Wavelet-based representations have consistently achieved comparable results to deep learning (Eickenberg et al., 2017; Zarka et al., 2020) with simple classifiers. More complex architectures, combining scattering with large neural networks (Patro & Agneeswaran, 2023), require an exorbitant amount of resources while offering little understanding of the representation.

### 4.1 TEXTURE CLASSIFICATION

Textural similarity is primarily attributed to Fourier phase, rather than magnitude, information with respect to human perception (Dong et al., 2017), making its preservation crucial in texture analysis. The KTH-TIPS benchmark (Fritz et al., 2004) remains a relevant testbed for examining the impact of higher-order, phase-preserving operators, particularly in domains where structural information drives variability—interactions encoded by the Solid Harmonic Wavelet Bispectrum. The dataset consists of 10 classes, each with 81 samples varying in scale, shear, and illumination.

We test our representations over 200 random train-test splits for different training set sizes, as done in (Sifre & Mallat, 2013). We train linear SVMs on our representations, which hold similar properties to the PCA classifier used in (Sifre & Mallat, 2013) over translation invariant (Mallat, 2012) and roto-translation invariant scattering (Sifre & Mallat, 2013). Enhanced roto-translation scattering refers to roto-translation scattering followed by a logarithmic nonlinearity, averaging over scales, and scale augmentations. Therefore, the most direct comparisons are between translation scattering, roto-translation scattering, and our proposed operators.

Table 1: Accuracy and standard deviation over 200 random training subsets with varying sizes of training set on KTH-TIPS

| Features | 5 samples | 20 samples | 40 samples |
|---|---|---|---|
| Translation scattering | $69.1 \pm 3.5$ | $94.8 \pm 1.3$ | $98.0 \pm 0.8$ |
| Roto-trans scattering | $69.5 \pm 3.6$ | $94.9 \pm 1.4$ | $98.3 \pm 0.9$ |
| SHWS | $86.9 \pm 0.8$ | $96.1 \pm 0.3$ | $98.5 \pm 0.4$ |
| SHWB | $87.7 \pm 1.7$ | $94.8 \pm 0.5$ | $96.9 \pm 0.3$ |
| SHWBic | $87.2 \pm 1.3$ | $96.5 \pm 0.6$ | $99.3 \pm 0.3$ |
| SHWSB | $\mathbf{88.6 \pm 2.5}$ | $95.1 \pm 0.3$ | $97.4 \pm 0.3$ |
| SHWSBic | $85.1 \pm 0.9$ | $97.0 \pm 0.7$ | $98.6 \pm 0.2$ |
| Enhanced roto-trans | $84.3 \pm 3.1$ | $\mathbf{98.3 \pm 0.9}$ | $\mathbf{99.4 \pm 0.4}$ |

SHWBic and SHWSBic outperform previous isolated scattering representations across all training sizes, demonstrating the added expressivity of bispectral representations in capturing higher-order nonlinearities. The improvement is most pronounced for the smallest training set ($n = 5$), where SHWSB achieves $88.6\%$ accuracy—an increase of almost 20 points compared to translation scattering. In this low-data regime, the bispectrum outperforms corresponding bicoherences, albeit with larger uncertainty (e.g., $\pm 2.5$ for SHWSB), suggesting that the additional information on total energy in frequency couplings introduces meaningful variability for classification.

As training sizes increase, this relationship shifts: bicoherence-based representations (SHWBic and SHWSBic) improve considerably, with 20-sample and 40-sample results reaching $96.5\%$ and $99.3\%$ accuracy, respectively, highlighting that patterns in the strength of phase coupling become more informative than raw energy. These results underscore the crucial role of phase-preserving, higher-order interactions for tasks driven by structural variability.

Solid Harmonic Wavelets consistently outperform Morlet scattering, emphasizing their capacity to encode textural information. While enhanced roto-translation scattering reaches high accuracy in larger training sets, SHWSBic achieves comparable performance with more certainty in intermediate

sizes, demonstrating the robustness of our operators across data regimes. Overall, these experiments highlight how the combination of bispectral interactions and roto-translation invariance produces rich, discriminative features that reduce uncertainty relative to earlier scattering-based approaches.

## 4.2 MEDICAL IMAGING

Deep learning in medical imaging is often limited by the availability of large-scale datasets (Zhou et al., 2021). Symmetry-aware wavelet operators mathematically impose key structural invariances, simplifying the variability within the representation and yielding meaningful features without large-scale optimisation. We evaluate this on MedMNIST (Yang et al., 2021; 2023), focusing on $28\times28$ images for fair comparison with existing deep learning baselines. MedMNIST presents distinct challenges: BreastMNIST requires detection of subtle shape irregularities, RetinaMNIST demands small vascular anomaly identification, and DermaMNIST relies on complex textural patterns—all benefiting from rotation-invariant analysis.

**BreastMNIST (780 samples)** is a binary classification task distinguishing benign from malignant tumors, which are discriminable based on symmetry and shape (Wei et al., 2020).

**RetinaMNIST (1,600 samples)** is an ordinal regression task grading diabetic retinopathy severity from 1–5, based on subtle vascular features (Porwal et al., 2020).

**DermaMNIST (10,015 samples)** is a seven-class classification task of pigmented skin lesions distinguished by asymmetry, border regularity, and color distribution.

For BreastMNIST, we use larger scales ($J \in [3, 4]$) to capture global symmetries. For RetinaMNIST and DermaMNIST, we focus on $J = 2$ for localized feature capture. Linear classifiers with $L_2$ regularization and singular value decomposition are used to isolate feature effectiveness.

Table 2: Accuracy over MedMNIST benchmark datasets. We performed an initial hyperparameter sweep of H-Net for each task over ring discretizations, number of filters, and angular frequencies. Other baselines are from Yang et al. (2021; 2023).

| Model | BreastMNIST | RetinaMNIST | DermaMNIST |
|---|---|---|---|
| ResNet-18 (Yang et al., 2021; 2023) | 86.3% | 52.5% | 73.5% |
| ResNet-50 (Yang et al., 2021; 2023) | 81.2% | 52.8% | 73.5% |
| auto-sklearn (Yang et al., 2021; 2023) | 80.3% | 51.5% | 71.9% |
| AutoKeras (Yang et al., 2021; 2023) | 83.1% | 50.3% | 74.9% |
| Google AutoML (Yang et al., 2021; 2023) | 86.1% | 53.1% | **76.8%** |
| H-Net (Worrall et al., 2017) | 84.4% | 48.7% | 72.4% |
| SHWS | 84.6% | 53.5% | 72.1% |
| SHWB | 78.2% | 52.8% | 69.7% |
| SHWBic | 82.7% | **54.5%** | 71.7% |
| SHWSB | 85.9% | 54.3% | 71.0% |
| SHWSBic | **86.5%** | 51.0% | 72.2% |

In data-scarce tasks (BreastMNIST, RetinaMNIST), our approaches surpass most baselines. SHWS-Bic achieves 86.5% accuracy on BreastMNIST, outperforming both ResNet-18 (86.3%), and H-Nets (84.4%). Our methods are particularly effective for medical images, which often contain Gaussian noise (to which the bispectrum is inherently insensitive), variable orientations (addressed by our roto-translation invariance), and multiscale features (captured by our wavelet decomposition). With larger datasets (DermaMNIST), our fixed-filter approaches remain competitive. Linear separability is evident: for instance, a Gaussian SVM on SHWSBic achieves 75.9% on DermaMNIST, outperforming all but one baseline.

SHWSBic outperforms the SHWBic on BreastMNIST, suggesting that a more sophisticated set of features is required to capture the variability of malignant tumors. Interference across scales is particularly important for that dataset. However, in RetinaMNIST key indicators are blurry blob-like structures with localised shape, making cross-scale interactions less discriminative. In this case, the scattering hierarchy in SHWSBic introduces features to minimal effect and SHWBic achieves the highest accuracy. Overall, these experiments highlight the advantage of built-in geometric invariances for data-efficient architectures.

### 4.3 ASTROPHYSICS

We study the application of SHWSBic in astrophysics. Galaxy mergers show complex features that have historically been challenging to identify using simple statistical measures (Pawlik et al., 2016; Conselice, 1997). Galaxy mergers exhibit complex, extended morphological structures that evolve over time and vary with mass ratio – precisely the type of multiscale geometric information our methods intend to capture.

We use a sample of galaxy mergers with mass ratios $\mu \geq -2$ within 2.5 Gyrs of a merger for a redshift up to 0.1 (following methodology of Avirett-Mackenzie et al. (2024)) derived from IllustrisTNG (Nelson et al., 2019). Data are extracted to match high spatial resolution from e.g. Hubble Space Telescope and do not have observational noise added. We downsample the initial dataset into two separate datasets for the regression of both time since last merger (n=18,791) and log merger mass ratio (n=16,870). Log merger mass ratio has mean $-1.10$ and standard deviation $0.54$, while time since last merger has mean 1.93 Gyr and standard deviation 1.06 Gyr. Images are cropped to $32\times$ the half stellar mass radius (as calculated from Nelson et al. (2019)) and resized to a resolution of $128 \times 128$. We optimise subject to mean-absolute-error (MAE). Experiments use $j = 4$ to capture larger, broader symmetries such as tidal tails.

Table 3: Results of merger feature regression on $128 \times 128$ resolution samples.

| Model | Log $\mu$ | | $\Delta t$ (Gyr) | |
|---|---|---|---|---|
| | MAE | RMSE | MAE | RMSE |
| Random predictor | 0.6479 | 0.7932 | 1.2306 | 1.5065 |
| Oriented scattering + linear | 0.4119 | 0.4953 | 0.8025 | 0.9604 |
| SHWS + linear | 0.3876 | 0.4701 | 0.7781 | 0.9360 |
| SHWS + KNN | 0.3539 | 0.4699 | 0.7264 | 0.9370 |
| **SHWSBic + KNN** | **0.3322** | **0.4498** | **0.6633** | 0.9139 |
| CNN | 0.3564 | 0.4586 | 0.7206 | 0.9406 |
| H-Net | 0.3581 | 0.4543 | 0.7345 | **0.9015** |

Table 3 shows that models trained on our operators effectively regress merger features. Best performing models achieve MAE of two-thirds the standard deviation for each target. We compare our approaches to a baseline random predictor, where predictions are sampled from the target range - demonstrating an almost 50% reduction in error rates. We observe an improvement using SHWS over oriented Morlet wavelets when input to linear models. We find significant improvements by training non-parametric models on our coefficients (KNN regressors with inverse $L_1$ distance weighting). This suggests that accurate prediction of merger features requires capturing not just global symmetries but also complex relationships between features at varying scales—precisely what SHWSBic encodes. The non-parametric approach enables the model to learn additional nonlinearities from our coefficients without imposing restrictive functional forms. SHWSBic achieves the best MAE across both tasks, though H-Nets obtain lower RMSE for time prediction. The difference between MAE and RMSE performance suggests that while our method achieves better typical-case accuracy, H-Nets may be more robust to outliers—possibly due to learned features adapting to rare morphological configurations. This highlights a characteristic trade-off: analytical features excel at capturing common structural patterns but may be less flexible for atypical cases without data-driven adaptation. Figure 1 shows true-predicted plots.

While previous studies have focused on binary merger classification (Avirett-Mackenzie et al., 2024; Walmsley et al., 2024; de Graaff et al., 2025), our approach pioneers regression on continuous merger features from morphological data. This is particularly valuable in astrophysics, where quantifying merger parameters typically requires extensive spectroscopic follow-up or access to simulation data. The strong relationship between true and predicted values (Figure 1) demonstrates that our wavelet-based approach effectively captures the complex structural signatures of merger dynamics, suggesting applications to large-scale observational surveys.

Figure 2 demonstrates the scaling of the predictive precision with respect to dataset size. All algorithms approach a similar MAE with increasing data points. The interesting observation is on

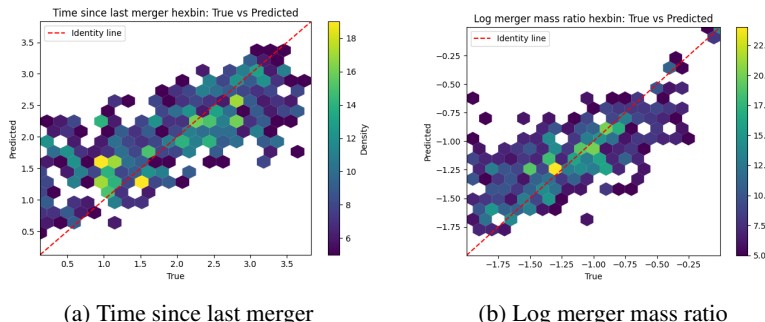

(a) Time since last merger          (b) Log merger mass ratio

Figure 1: True-predicted hexbin plots using K-Nearest-Neighbours regressor with SHWSBic input. Hexbins with more than 5 samples are shown to highlight underlying trends and remove outliers.

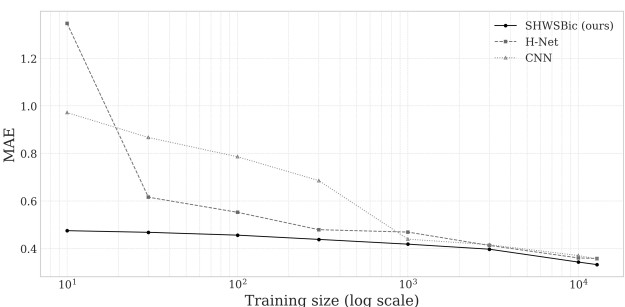

Figure 2: Comparison of Mean Absolute Error for log mass ratio regression with increasing training set sizes (log scale). SHWSBic features are used with a KNN regressor.

the low data regime, where our method substantially outperforms Deep Learning and Geometric Deep Learning baselines. The low-data regime is interesting due to the rarity of galaxy mergers in both simulations and even more in real observational data. Additionally, Figure 2 supports the case that the SHWSBic space is a natural space for comparisons of galaxies. Hybrid approaches combining wavelet features with trainable layers could further improve performance (Zarka et al., 2020; Oyallon et al., 2014); however, naively minimizing errors could obscure the contribution of the representation relative to the supervised learning algorithm.

## 5 CONCLUSION

We introduce new operators that generate roto-translation invariant features for supervised learning over image-based tasks. The **Solid Harmonic Wavelet Bispectrum (SHWB)** captures multi-scale interactions between feature types, while preserving phase information. Building on this, the **Solid Harmonic Wavelet Scattering Bispectrum (SHWSB)** combines properties of both scattering networks and the solid harmonic wavelet bispectrum, enabling the encoding of higher-order nonlinearities while retaining critical phase information.

Experiments with simple linear models demonstrate that these representations are increasingly discriminative and perform competitively with deep learning in limited-data settings. For textures, SHWB and SHWSB capture structural variability and higher-order interactions, outperforming standard scattering representations. In medical imaging, our operators effectively linearize decision boundaries and achieve state-of-the-art or competitive accuracy on small datasets. In astrophysics, SHWSBic successfully predicts merger features, particularly in low-data regimes, highlighting the natural expressivity of phase-preserving, multi-scale operators.

A limitation of our study is the lack of evaluation on large-scale datasets such as ImageNet. While such benchmarks are standard for deep learning, they may not accurately reflect the advantages of our operators. Hybrid approaches combining wavelet features with trainable layers (Oyallon et al.,

2014; Patro & Agneeswaran, 2023) could further enhance performance, but their inclusion would obscure the contribution of the operator itself and add significant computational overhead.

Overall, our work highlights the crucial role of phase-preserving, higher-order interactions in extracting discriminative features. The proposed operators provide interpretable, mathematically grounded, and data-efficient representations well-suited for tasks where symmetry and multi-scale structure are central.

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

## A  TECHNICAL APPENDICES AND SUPPLEMENTARY MATERIAL

### A.1  DATASETS

We plot visualisations of MedMNIST samples in figure 3. In each task, relevant occur at varying scales. For example, BreastMNIST relies on broader symmetries such as regularity of the tumour.

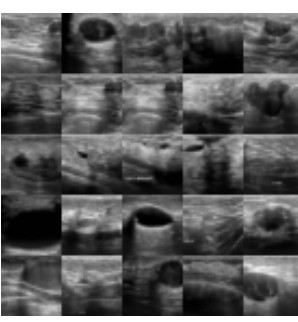 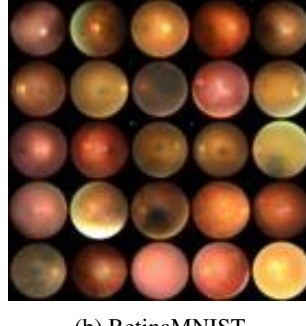 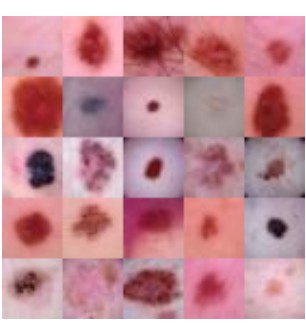

(a) BreastMNIST                  (b) RetinaMNIST                  (c) DermaMNIST

Figure 3: MedMNIST dataset samples

In figure 4 we present a selection of random samples from our galaxy dataset. Evidently, there is no observational noise. Similarly, one can observe the presence of merger features; for example, the top-left sample depicts tidal tails.

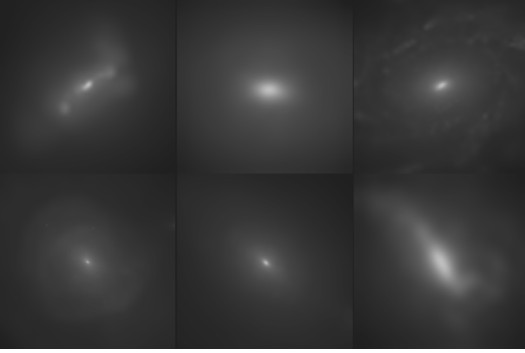

Figure 4: Galaxy merger samples.

Finally, samples from KTH-TIPS (figure 5) are comparatively simpler. Classes are distinct, and discrimination is subject to solely textural patterns at varying scales.

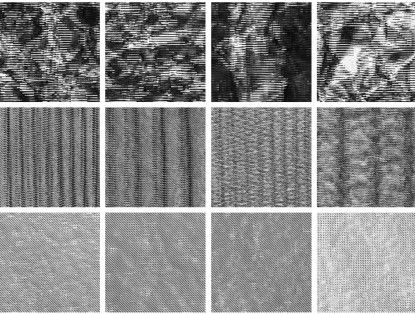

Figure 5: Samples from KTH-TIPS. Each row is a different class.

## A.2 OPERATORS

Figure 6a shows solid harmonic wavelets over varying scales (horizontally) and angular frequencies (vertically). Figure 6b shows the components that build up solid harmonic wavelets from Eq. 22. The Gaussian window, $\frac{1}{\sqrt{(2\pi\sigma^2)}}e^{-r^2/2\sigma^2}$ plotted on the top left is used to localise the wavelet support, while the radial component, $r^\ell$ top right, removes a singularity at 0. Finally, the complex frequency components, $e^{i\ell\varphi}$ with real and imaginary plotted at bottom left and right respectively, span the angular coordinate of the wavelet.

$$\psi_{\sigma,\ell}(r,\varphi) = \frac{1}{\sqrt{(2\pi\sigma^2)}}e^{-\frac{r^2}{2\sigma^2}}r^\ell e^{i\ell\varphi} \tag{22}$$

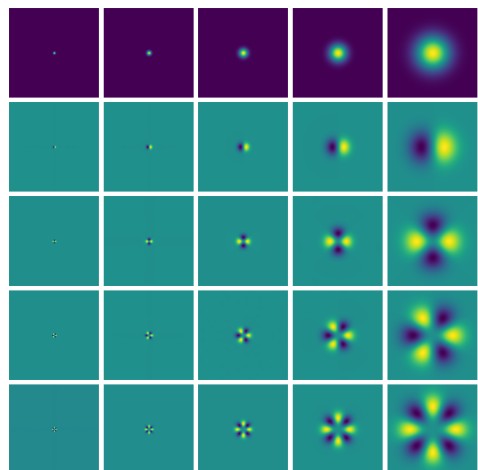

(a) Real parts of solid harmonic filter bank. J increases left-to-right. L increases top-to-bottom.

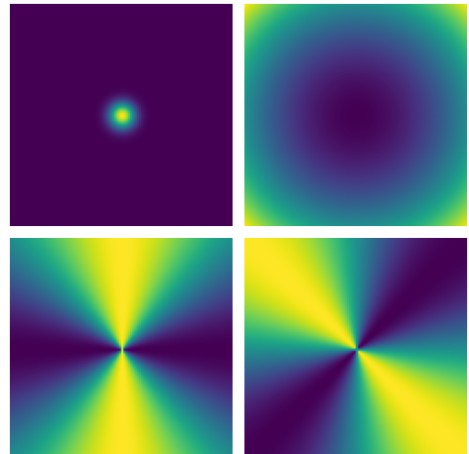

(b) Decomposition of wavelet. Top left to bottom right: Gaussian window, radial, real and imaginary angular components.

Figure 6: Solid harmonic wavelets in 2D

Figure 7 compares wavelet responses for both Morlet (Andreux et al., 2018) and Solid Harmonic filters. While Morlet wavelets effectively extract prominent features, such as edges, solid harmonic wavelets excel in encoding textural information – in addition to edges.

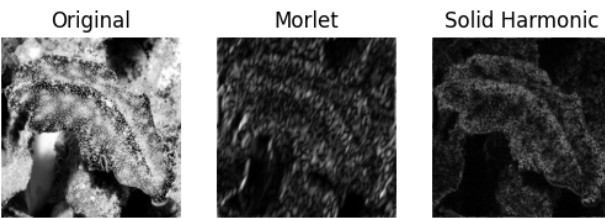

Figure 7: Wavelet transform responses using Morlet and Solid Harmonic wavelets

We plot how extracted features from the solid harmonic wavelet transform vary with solid harmonic wavelet parametrisation in figure 8. By increasing $\ell$, the features extracted by the solid harmonic wavelet transform transition from edges to texture, such that using dilated wavelets one could encode both textures and edges across scales. Thus, solid harmonic wavelet responses more effectively decouple these feature types. One can reason that since increasing the angular frequency produces more sinusoidal oscillations, solid harmonic wavelet transforms with increasing $\ell$ can capture finer-scale variations in texture as a result of increased sensitivity to directional variations in the input

signal. Simultaneously, increasing the scale $j$ captures coarser, higher-level features. To this extent, by building a filter bank of dilated solid harmonic wavelets one can produce a complete representation which encompasses edges and textures of varying granularity. Contrasting to Morlet wavelet transforms in figure 7, we demonstrate that solid harmonic wavelet transforms capture an increasingly diverse collection of feature representations.

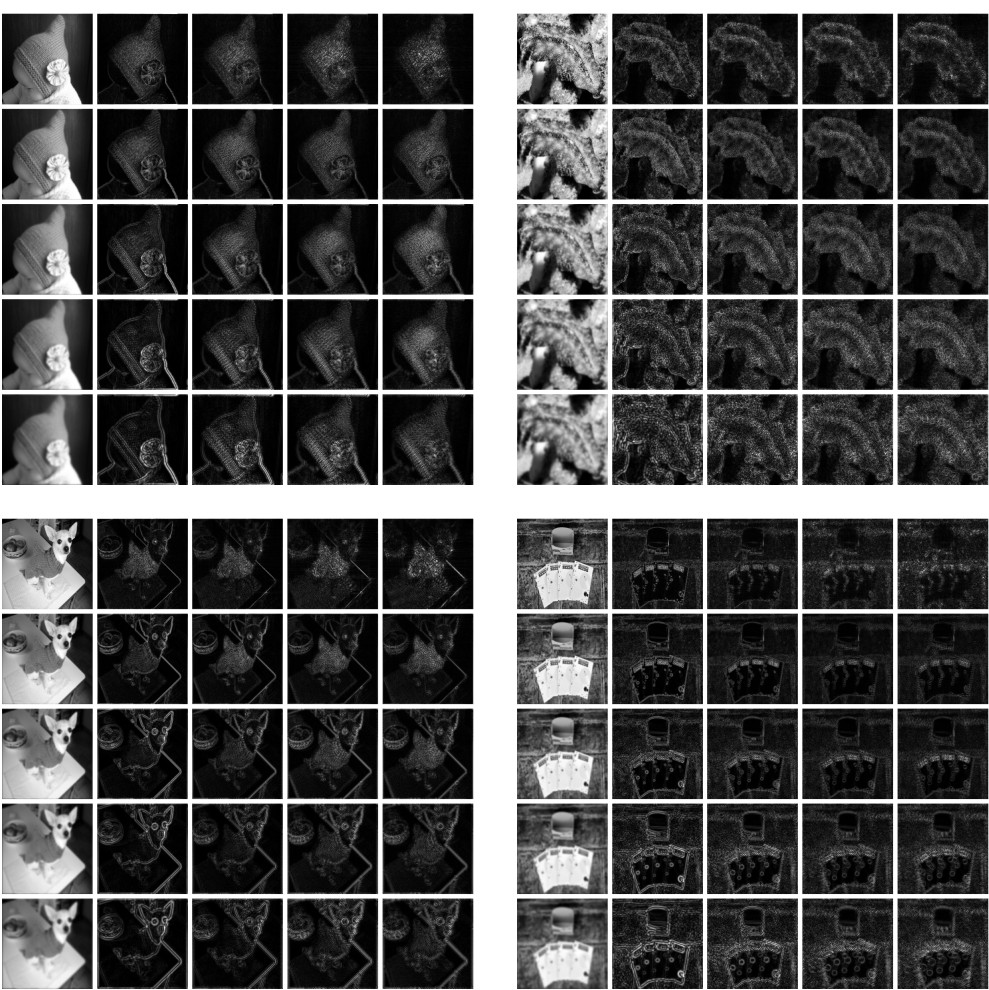

Figure 8: Effect of $\ell$ and $j$ parameters on features produced by solid harmonic wavelet transform. Left-to-right: $\ell$ increases from 0 (Gaussian) to 10 in steps of 2. Top-to-bottom: $j$ increases.

In figure 9 we plot the magnitude for $L_1$ pooling of SHWBic coefficients. One can observe similarity between samples belonging to the same class (babies).

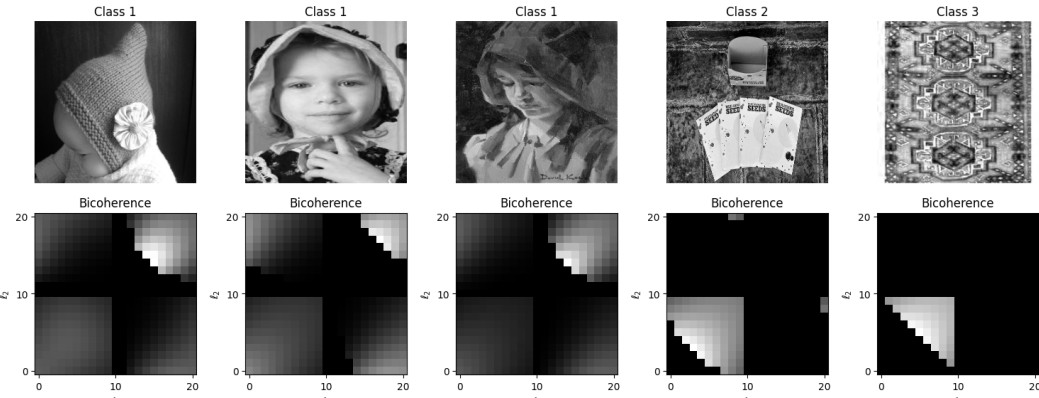

Figure 9: Plots for SHWBic $L_1$ pooling magnitude for different ImageNet classes. Note intra-class similarity.

### A.3 BISPECTRUM COVARIANCE

This appendix establishes that SHWB and SHWSB are covariant to roto-translations under the special Euclidean group SE(2), and that integration over spatial frequencies yields invariants. To establish the covariance of the solid harmonic wavelet bispectrum (SHWB) under rotations in Eq. 23, we begin by considering the rotational behavior of the solid harmonic wavelet transform. For a planar rotation $R_\gamma$, the wavelet coefficients transform as shown in Eq. 8:

$$W_{j,\ell}(R_\gamma x) = e^{i\ell\gamma} R_\gamma W_{j,\ell}(x),$$

where $R_\gamma$ denotes the spatial rotation operator acting on the coordinates, and the phase term $e^{i\ell\gamma}$ arises from the equivariance of solid harmonics under rotation of the input signal.

We substitute Equation 8 into the definition of the SHWB:

$$\text{SHWB}[x, j, \ell_1, \ell_2] = W_{j,\ell_1}(x) \cdot W_{j,\ell_2}(x) \cdot \overline{W_{j,\ell_1+\ell_2}(x)},$$

Applying a rotation $R_\gamma$ to the input $x$ and using Equation 8, we have:

$$\begin{aligned}
\text{SHWB}[R_\gamma x, j, \ell_1, \ell_2] &= W_{j,\ell_1}(R_\gamma x) \cdot W_{j,\ell_2}(R_\gamma x) \cdot \overline{W_{j,\ell_1+\ell_2}(R_\gamma x)} \\
&= e^{i\ell_1\gamma} R_\gamma W_{j,\ell_1}(x) \cdot e^{i\ell_2\gamma} R_\gamma W_{j,\ell_2}(x) \cdot \overline{e^{i(\ell_1+\ell_2)\gamma} R_\gamma W_{j,\ell_1+\ell_2}(x)} \\
&= e^{i(\ell_1+\ell_2-(\ell_1+\ell_2))\gamma} R_\gamma \left( W_{j,\ell_1}(x) \cdot W_{j,\ell_2}(x) \cdot \overline{W_{j,\ell_1+\ell_2}(x)} \right) \\
&= R_\gamma \text{SHWB}[x, j, \ell_1, \ell_2].
\end{aligned} \tag{23}$$

Thus, the SHWB is *covariant* to rotations. Crucially, this covariance hinges on the angular momentum conservation condition $\ell_3 = \ell_1 + \ell_2$, which ensures that the global phase factor $e^{i(\ell_1+\ell_2-\ell_3)\gamma}$ cancels. This cancellation is what distinguishes the SHWB from classical wavelet scattering: the phase information is preserved without requiring a modulus to enforce rotational invariance.

Since the underlying wavelet convolutions are also covariant to translations Mallat (2012), the SHWB inherits translation covariance. Full *roto-translation invariance* is then achieved by applying a suitable *reduction operator*, such as spatial averaging over the domain.

This formalism justifies the invariance properties of SHWB and illustrates how higher-order interactions can retain geometric structure without sacrificing phase information, offering a more expressive alternative to classical scattering transforms.

Finally, we note that the solid harmonic wavelets $\psi_{\ell,j}$, producing the responses $W_{j,\ell}(x)$, are orthogonal across different scales $j$ and angular frequencies $\ell$. As a result, the SHWB generates a set

of largely independent invariants. This orthogonality ensures that while irrelevant variations (e.g., due to transformations) are removed, the essential discriminative information in the signal is largely preserved. Consequently, the SHWB forms a rich invariant representation that captures higher-order geometric structure without discarding informative phase content.

