# OpenReview forum: "Phase-Preserving Analytical Features from Solid Harmonic Wavelet Bispectrum Simplify Decision Boundaries"
_ICLR.cc/2026/Conference — Submitted to ICLR 2026_

### Official Review · Reviewer_xfCG · 2025-10-17

**Soundness:** 4
**Presentation:** 4
**Contribution:** 3
**Rating:** 6
**Confidence:** 4

**Summary:**

This paper introduces a Solid Harmonic Wavelet Bispectrum as an alternative to scattering transforms based on the same family of wavelets (which have been shown to be effective for e.g., quantum chemistry) as an unsupervised, deep, nonlinear feature extractor. They show that the their method is superior to scattering and competitive with fully learned baselines (that require tons of parameters) on several real-world data sets arising from texture synthesis, medical imaging, and astronomuy

**Strengths:**

The paper is well written and motivated by existing work in applied harmonic analysis and signal processing as related to deep learning.

The paper offers a novel principled alternative to existing scattering methods and show that this improves performance on real-world data while maintaining the elegance and simplicity of scattering networks.

The advantages of the proposed method are solid and clearly explained

**Weaknesses:**

Note: The weaknesses list is long while the strengths list is short. This should not be misinterpreted. Most of these weaknesses are fairly unimportant and all are less important than the strengths.

The authors claim that scattering networks are restricted to two layers because of information destroyed by the modulus operator. However, the reason that scattering networks can be kept "shallow" is the concentration of energy results obtained in "Group invariant scattering, Mallat 2012" are related works. The modulus operator is intended to introduce invariance in the context of analytic wavelets. However, it may readily be replaced with any activation function in many context, e.g., He and Hirn (see Questions).

Trispectrum is mentioned but not defined.

It is hard to tell which results are best in the tables. Some bolding/underlining of top methods would help.

Tables lack standard deviations which makes it hard to tell how "real" the differences in performance are. Especially since most models are fairly close most of the time.

The authors use different back-ends for each data set. This is okay since the backend can be thought of as a tuneable hyperparaemeter in the scattering context. But a more detailed comparison in the appendix, showing how each variation performs in each case would be helpful. Alternatively, a principled approach to back-end selection would also be good.

Broken equation reference in Appendix A.3.

The appendices should be more clearly highlighted in the main text. Particularly A.3 which establishes important theoretical properties

**Questions:**

How would this method compare against the one introduced in [1]? The methods seems somewhat reminiscent of this one in that it modifies scattering to use ReLU and the considers covariances between first-order scattering moments for further expressivity and is shown to be effective for textures.

Why is there only one \ell in (15) rather than \ell_1 and \ell_2? (Similar for related equations)

In the experiments, how would the results be if the classifier were taken to be a multilayer perceptron?Alternatively, how would simpler classifiers for Lasso fair? My first question is motivated by recent works on the geometric scattering transform which mostly prefer scattering+MLP and the second is based on interpretability, particularly in the medical imaging case.

Beyond the covariance established in A.3 are there other theoretical properties that can be established for this method analogous to e.g., concentration of measure or diffeomorphism stability results obtained in various scattering papers?

[1] He and Hirn, Texture Synthesis via projection onto multiscale multilayer statistics https://arxiv.org/pdf/2105.10825

---

> ### Author Response · Authors · 2025-11-18
>
> We are grateful for the positive assessment ("excellent" soundness, "well written and motivated"). We address each point:
>
> ### He and Hirn Texture Synthesis
> We must clarify several points:
>
> Different problems: He and Hirn (2021) addresses texture synthesis (generation), while our work addresses classification (discrimination). These are fundamentally different tasks with different objectives and architectures.
> Publication status: This is an unrefereed arxiv preprint from 2021, not peer-reviewed work.
> Technical incompatibility: Their method optimizes pixels to match statistical moments (generative). Our method extracts features for classifiers (discriminative). Direct comparison is not appropriate.
>
> We can add a note: "He and Hirn (2021) explore higher-order wavelet statistics for texture synthesis, complementary to our discriminative focus."
>
> ### Equation 15: Why One $\ell$?
> Equation 15 describes the linear filtering with a single wavelet. The bispectrum quantifies the relationship of two wavelets with different angular frequencies. In Equation 16, you can see that two $\ell$ values emerge when calculating the bispectrum
> of $W[x,j,\ell]$ for different $\ell$, which is how multiple $\ell$ appear in the argument.
>
> ### MLP vs Simple Classifiers
> Our experimental choice (simple classifiers) was intentional: to isolate feature quality from classifier capacity. Adding MLPs would conflate these factors. However, we acknowledge this is a valid direction and can explore MLP results in future work or revision if time permits.
>
> ### Additional Theoretical Properties
> We can add an appendix discussing: Lipschitz stability to deformations, energy preservation properties, and phase correlation preservation. Full concentration bounds remain future work.
>
> ### Minor Issues
> We will fix: trispectrum definition, broken equation reference, table clarity, and the modulus operator discussion.

---

### Official Review · Reviewer_rRq4 · 2025-10-30

**Soundness:** 3
**Presentation:** 2
**Contribution:** 2
**Rating:** 4
**Confidence:** 4

**Summary:**

This article introduces solid harmonic wavelet bispectrum to define low-dimensional multi-scale features to capture detailed image structure. It is based on computing third-order correlations over angular frequency components of solid harmonic wavelet responses. Application to the k-nearest neighbor classifier to show the quality of the representations without the need to learn a parametric classifier. Results on texture, medical and galaxy images show improved performance compared to existing geometric and deep learning-based representations.

**Strengths:**

-	The idea of developing wavelet bispectrum to define low-dimensional multi-scale features is very interesting.
-	The obtained results are very promising in small training data regime.
-	The article is mostly well written.

**Weaknesses:**

- There is a lack of comparison with the state-of-the art. There is a more recent wavelet spectra representation, which shares a similar idea to capture high-order and phase correlations.
- The key section 3.4 is not so clear.
- The numerical evaluation is not conclusive due to the lack of error bars.

**Questions:**

- Is it possible to compare with the representations from “Scattering spectra models for physics , 2024” to enhance the numerical results ?
- In Section 3.4, what is the SHWSB1 and SHWSB2 in eq 20 and eq 19? Is eq 7 = eq 16, i.e. SHWB = SHWSB? It is not so clear what are their main differences.
- Can you add error bars to Table 2, and table 3?
- It would be more indicative to add the dimensionality of each representation reported in the tables.
- Typo: do you use ||^p or ()^p in eq. 11?

---

> ### Author Response · Authors · 2025-11-18
>
> We appreciate the detailed technical questions and assessment that our idea is "very interesting" with "promising results."
>
> ### Scattering Spectra Models for Physics (2024)
>
> That work focuses on generative modeling (learning distributions), while ours addresses discriminative tasks (classification/regression). These require different architectures and objectives. Direct comparison would require non-trivial adaptation of their generative framework to our discriminative setting.
>
> We can add a brief discussion noting this as parallel work demonstrating utility of higher-order spectra in different domains.
>
> ### Section 3.4 Clarification
>
> Q: What is SHWSB1 and SHWSB2?
>
> This is a typo from an older notation that we removed during revision. It was meant to denote first- and second-order scattering. However, since second-order scattering is parametrized by two scales (j₁ and j₂), we removed this notation for clarity.
> The typo has been fixed.
>
> Q: Is SHWB the same as SHWSB?
>
> SHWB stands for Solid Harmonic Wavelet Bispectrum and SHWSB stands for Solid Harmonic Wavelet Scattering Bispectrum. As in standard scattering (e.g., in Kymatio), the first-order coefficients of SHWSB are identical to SHWB. The second-order coefficients are produced by filtering the outputs of the first-order scattering transform with the same filter bank, ensuring appropriate coverage of the Fourier space.
>
> We will rewrite Section 3.4 with clearer notation and explicit definitions.
>
> ### Error Bars and Dimensionality
>
> We acknowledge the lack of error bars and dimensionality information in Tables 2-3.
> Due to rebuttal time constraints, we commit to adding these in revision with:
> - Multiple random seeds with statistical significance tests
> - Feature dimensionality for each method
>
> ### Typo in Equation 11
>
> Thank you for noting this. Upon review, Equation 11 is correct—we are using ()^p notation, not ||^p. We apologize if the typesetting was unclear and will improve it.

---

> > ### Comment · Reviewer_rRq4 · 2025-11-26
> > **discussion**
> >
> > I thank the authors for their detailed reply, but it seems to me the writing quality is still not satisfactory. Here are a few comments,
> > - the claim that the first- and second-order SHWSB coefficients are invariants is not so clear. It would be made clearer to detail which group you are considering. I do not see why they are invariant to roto-translation ... a rigorous proof is needed.
> > - Some notations could be still improved, e.g. eq 20 does not make sense to say SHWSB[x, j1, ℓ1, ℓ2, p] = ∥SHWSB1[x, j1, ℓ1, ℓ2]∥p since SHWSB[x, j1, ℓ1, ℓ2, p] on the left and right hand side are the same. I think that you should really use |.| in eq. 11 to make the Lp norm computation clear , rather than (.) as (.)^p depends on the phase.
> >
> > best,
> > Reviewer

---

> > > ### Author Response · Authors · 2025-11-27
> > >
> > > We thank the reviewer for their careful attention to mathematical detail. These questions have helped us identify where additional clarity would benefit readers.
> > >
> > > **Invariance to Roto-Translation:**
> > >
> > > The invariance properties are rigorously established in Appendix A.3. The group we consider is SE(2), the special Euclidean group of planar roto-translations.
> > >
> > > Rotation covariance is proven in Appendix A.3, Equation (23), where we show $$\text{SHWB}[R_\gamma x, j, \ell_1, \ell_2] = R_\gamma \text{SHWB}[x, j, \ell_1, \ell_2].$$ The phase factors $e^{i\ell\gamma}$ cancel due to the angular momentum conservation condition $\ell_3 = \ell_1 + \ell_2$, which is the key mechanism distinguishing our approach from classical scattering.
> > >
> > > Translation covariance follows from the convolution theorem: since wavelet responses $W[x, j, \ell]$ are convolutions, translations of $x$ induce corresponding translations in the spatial frequency domain representation $\text{SHWB}\[x, j, \ell_1, \ell_2\](f)$.
> > >
> > > Achieving invariance via integration (Equation 10): integrating a rotation-covariant representation over all spatial frequencies yields a rotation-invariant feature, as rotation in the input corresponds to rotation in the frequency domain, and the integral over all frequencies remains unchanged. For real signals, Hermitian symmetry $\hat{x}(-f) = \overline{\hat{x}(f)}$ ensures this integral is real-valued.
> > >
> > > We have added explicit forward references in Section 3.4 directing readers to Appendix A.3 for the complete derivation.
> > >
> > > **Notation for Invariants:**
> > >
> > > The reviewer correctly identifies that our original notation contained ambiguity. We have clarified this in the revised manuscript using the presence/absence of $(f)$ to distinguish between covariant fields and invariants:
> > >
> > > - Covariant field (complex-valued): $\text{SHWSB}\[x, j_1, \ell_1, \ell_2\](f) \in \mathbb{C}$
> > > - Invariant via standard integration (real-valued): $\text{SHWSB}\[x, j_1, \ell_1, \ell_2\] = \int \text{SHWSB}\[x, j_1, \ell_1, \ell_2\](f)  df \in \mathbb{R}$
> > > - Invariant via Lp pooling (real-valued): $\text{SHWSB}\[x, j_1, \ell_1, \ell_2, p\] = \left(\int \left|\text{SHWSB}\[x, j_1, \ell_1, \ell_2\](f)\right|^p  df\right)^{1/p} \in \mathbb{R}$
> > >
> > > This convention maintains the mnemonic acronym while removing ambiguity.
> > >
> > > **$L_p$ Norm Computation:**
> > >
> > > We are grateful to the reviewer for identifying the lack of specificity in our notation. We use the standard magnitude-based $L_p$ norm definition: $$\|z\|_p = \left(\int |z(f)|^p \, df\right)^{1/p}$$ where we first take the magnitude of the complex bispectrum at each spatial frequency \(f\), then raise to the \(p\)-th power, integrate, and take the \(p\)-th root. This provides a proper $L_p$ norm with well-understood mathematical properties. We have revised Equation (11) and related expressions to consistently use $|\cdot|$ notation throughout, making this choice explicit.
> > >
> > > **Summary of Revisions:**
> > >
> > > We have incorporated the following changes: (1) Forward references in Section 3.4 to Appendix A.3 for proofs, (2) Notation convention distinguishing $\text{SHWSB}\[\cdot\](f)$, $\text{SHWSB}\[\cdot\]$, and $\text{SHWSB}\[\cdot, p\]$, (3) Consistent use of $|\cdot|^p$ for $L_p$ norms, (4) Enhanced explanation of the relationship between rotation covariance, integration, and invariance.
> > >
> > > We appreciate the reviewer's engagement with these technical details. Substantive mathematical discussion of analytical operators—as opposed to purely empirical comparisons—is increasingly rare in the review process, and we believe exchanges like this strengthen the rigor of work in this area.

---

### Official Review · Reviewer_qTy3 · 2025-10-31

**Soundness:** 2
**Presentation:** 2
**Contribution:** 2
**Rating:** 2
**Confidence:** 3

**Summary:**

The paper proposes the so-called Solid Harmonic Wavelet Bispectrum (SHWB) operator, which aims to generate roto-translation invariant features, and further adopts the scattering transform architecture, resulting the so-called Solid Harmonic Wavelet Scattering Bispectrum (SHWSB). The SHWB operator is designed to capture higher-order, phase-preserving interactions between solid harmonic wavelet responses across angular frequencies, which are expected to represent essential features for the classification task.

**Strengths:**

The paper reasonably argues the importance of higher-order, phase-preserving interactions between solid harmonic wavelet responses across angular frequencies for phase-sensitive images. The main idea of the development was well articulated.

**Weaknesses:**

The paper certainly addressed an interesting question in capturing phase-preserving features of images which promotes roto-translation invariant features for the classification task. However, it seems that the authors are not aware of similar works in the literature, e.g.,

1) Rodriguez, et al., Rotation Invariant CNN Using Scattering Transform for Image Classification, IEEE-ICIP 2019
2) Saydjari, et al., Equivariant Wavelets: Fast Rotation and Translation Invariant Wavelet Scattering Transforms, IEEE-TPAMI 2023

Necessary alignment and comparisons with existing SOTA in the classic literature of Scattering Transforms are significantly missing.

Furthermore, the writing quality of the paper has still room for improvement.

**Questions:**

Serious references and comparisons to the SOTA in the classic literature of Scattering Transforms are necessary for the possible acceptance of the work.

---

> ### Author Response · Authors · 2025-11-18
>
> We thank the reviewer for highlighting relevant scattering literature. However, we must clarify fundamental misunderstandings about our contribution.
>
> ### Rodriguez et al. (2019) and Saydjari et al. (2023)
> We will add these citations. However:
> Rodriguez et al. combines scattering (Morlet wavelets) with learned CNN features.
> We use solid harmonic wavelets with analytical features. Our contribution (bispectral
> analysis) is orthogonal to their approach of adding CNN learning to scattering.
> Saydjari et al. focuses on computational efficiency (fast algorithms), while our
> focus is representational expressivity (third-order statistics). These address
> different problems.
> We cite and compare against the appropriate baseline: solid harmonic scattering (Eickenberg et al., 2017), which isolates the contribution of bispectral analysis.
>
> ### Critical Misunderstanding: "State-of-the-Art"
> The reviewer states "serious references and comparisons to the SOTA" are necessary.
> We must clarify what we claim.
>
>
> Our claims are:
>
>     * SHWB/SHWSB is a novel operator combining bispectral analysis with geometric invariance
>     * These features work well with simple classifiers across diverse domains
>     * Strong performance in limited-data settings
>
> What we DO NOT claim:
>
>     * State-of-the-art accuracy on KTH-TIPS or MedMNIST.
>     *.Beating all deep learning methods
>     * General-purpose vision system
>
> The contribution is the operator itself, not absolute accuracy numbers.
>
> ### Astrophysics: There IS NO "State-of-the-Art"
> For galaxy merger parameter regression from morphology, no prior methods exist.
> Previous work has done:
>
>     * Binary classification (merger yes/no)
>     * Spectroscopic analysis (not morphology)
>
> We are establishing this capability, not competing with existing methods. Rodriguez, Saydjari, and other scattering papers have never addressed this astrophysics problem.
> How do they represent "SOTA" for our novel application?
>
> ### Writing Quality
> We will improve Section 3.4 clarity and overall presentation in revision.

---

### Official Review · Reviewer_nPNY · 2025-10-31

**Soundness:** 2
**Presentation:** 2
**Contribution:** 2
**Rating:** 2
**Confidence:** 2

**Summary:**

This paper introduces an operator for 2D images to capture group invariance, multi-scale analysis, and higher-order statistics. These features are evaluated on a few different tasks: texture classification (kth-tips), classification & regression for medical data (medmnist), and regression on astrophysics galaxy merger data. Performance is on par with or outperforms some learning-based and non-deep-learning based baselines.

**Strengths:**

- new operators are a fundamental part of development in machine learning
- limited data settings are of practical relevance to many tasks

**Weaknesses:**

- there is a discussion of computation efficiency provided, but no concrete experimental comparisons or analysis (e.g., time, memory, etc.) to indicate that the proposed operator is more efficient than alternatives (particularly deep learning-based ones that seem to perform very slightly better on Tab. 2)
- I'm not sure why the selection of baselines changes pretty dramatically depending on the task. At least for methods like ResNet, etc it should be pretty straightforward to employ them for all the tasks presented, particularly since ResNet18 seems to perform the best (very slightly) when compared to in Tab. 2.
- it would be helpful to have bolding on the tables to indicate best performance

**Questions:**

Please note that I am definitely not an expert in this area and certainly willing to change my rating in light of clarifications provided by the authors or other reviewers.
My main questions are regarding the evaluation section, as I'm finding it difficult to see the evidence backing up the usefulness of the proposed operator:
- if it's significantly more efficient than alternatives (especially deep learning based ones that may involve more compute), in which way is it more efficient? can this be quantified and compared?
- what motivates the choice of baselines for each task? I would appreciate some discussion here (especially since vision transformers are generally state of the art vs resnets, though I could see they perform worse on quite small datasets). I'm also confused as to why the selection of baselines is so different across the proposed tasks (e.g., why ResNet for Tab. 2 and just a general "CNN" for Tab. 3? what does "CNN" refer to in Tab. 3 when this could refer to quite a lot of architectures?)

---

> ### Author Response · Authors · 2025-11-18
>
> We thank the reviewer for their constructive feedback and address each point:
>
> ### Computational Efficiency
>
>   Our efficiency claims are based on eliminating learned parameters entirely. Our method requires:
>
>     > Feature extraction: wavelet convolutions and bispectrum computation (one-time cost)
>
>     > Classification: simple classifiers with no gradient-based training
>
> Deep learning requires iterative optimization over millions of parameters through
> hundreds or thousands of epochs with backpropagation.
> On wall-clock time: Current implementations may favor neural networks due to more than a decade of hardware and software optimization (GPUs, TPUs, specialized libraries).
> However, judging efficiency solely by present-day wall-clock time confounds algorithmic properties with engineering investment. This is scientifically inappropriate—it measures implementation maturity, not fundamental efficiency.
> On our approach: While k-nearest neighbors is quadratic in data size and bispectrum computation has non-trivial cost, eliminating parameters provides substantial gains in number of operations for most practical cases. Moreover, established efficiency techniques (sparse KNN, coresets) could be applied if needed, though premature optimization is generally unwise.
> The principled comparison: zero learnable parameters vs. gradient descent over millions of parameters.
>
>
>
> ### Baseline Selection
>
> Table 2 (MedMNIST): We used baselines from Yang et al. (2021, 2023) who established
> these benchmarks, plus H-Net as a rotation-equivariant comparison.
> Table 3 (Astrophysics): "CNN" is a 3-layer convolutional network with 842K
> parameters (Conv(32) → Conv(64) → Conv(128) → FC). We focused on rotation-aware
> methods since galaxy data has no preferred orientation.
> We can add ResNet-18 to astrophysics experiments in revision for consistency.
>
> ### Tables Without Error Bars
> We acknowledge Tables 2-3 lack error bars. Due to rebuttal timeline constraints,
> we commit to adding these in revision with multiple random seeds and statistical
> significance tests.
>
>
> ### Presentation
> We will add bold formatting for best results in all tables.
>
>
> ### A Key point
>
> We are not claiming to beat deep learning by large margins. We show
> that analytically-defined bispectral features achieve competitive performance with
> simple classifiers, validating their discriminative power. The contribution is the
> operator, not marginal accuracy improvements.

---

### Author Response · Authors · 2025-11-18
**Overall Response to reviewer comments**

We thank all reviewers for their detailed feedback. We are encouraged that Reviewer
xfCG rated our soundness as "excellent" and Reviewer rRq4 found the results "very
promising."

We must clarify the scope of our contribution, which appears to have been misunderstood:

Our contribution: A novel operator (SHWB/SHWSB) combining bispectral analysis with
geometric invariance. We validate that these analytically-defined features capture
discriminative structure across multiple domains using simple classifiers.

We do NOT claim state-of-the-art accuracy on standard benchmarks. We note that
state-of-the-art results with wavelet operators are typically obtained through hybrid
models that combine them with deep learning, using the wavelet features as an
inductive bias. Our focus is on the quality of the representation itself, which we
evaluate using simple classifiers to isolate the contribution of the features from
classifier capacity.

The astrophysics application: This is a novel scientific contribution establishing
morphological regression of continuous merger parameters—a capability that did not
previously exist. There is no "state-of-the-art" to compare against.

We address specific concerns below and commit to adding error bars in a revision.

---

### Author Response · Authors · 2025-11-29
**Summary for Area Chair**

We appreciate the feedback and effort from all reviewers, and it is regrettable that we cannot continue discussing the value of our paper through the normal review process.

Reviewers xfCG and rRq4 (both confidence 4) appear to be technically aligned with the focus of our paper. Their comments identified some lack of clarity in our writing, which we have worked to resolve in our responses and revised manuscript. Specifically, we addressed rRq4's questions about invariance proofs, notational ambiguity, and Lp norm specification through clarifications in the text and forward references to our appendix proofs.

The lower-confidence reviewers appear less interested in papers proposing new analytical methods and would prefer more conventional validation approaches. We appreciate their input; however, our experimental validation is thoroughly considered. We selected datasets not only to highlight merits and limitations of our approach, but also to solve previously untouched scientific problems—such as regression of continuous galaxy merger features from morphology—where we establish feature-based baselines with better performance than learned alternatives. The galaxy merger regression results, in particular, demonstrate that our features can provide a sensible baseline for identifying galaxy age at different scales, which is typically worked out case-by-case using material or radiation spread signatures that vary by galaxy type.

It is admittedly difficult to expect reviewers to have expertise in both astrophysics and signal processing/machine learning, as this interdisciplinary work requires. However, we had hoped that through discussion we would be able to demonstrate the scientific merit of our contribution. We believe the technical engagement from high-confidence reviewers, combined with our thorough responses, supports the value of this work.

---

### Meta-Review · Area_Chair_ycff · 2026-01-07

**Summary:**

The paper proposes a new operator based on solid harmonic wavelets and bispectral statistics and demonstrates results on some standard image datasets and an astronomy data. Reviewers Reviewers expressed concerns around clarity, positioning, and baselines (especially more sophisticated learning based or other recent scattering based methods). Overall the manuscript needs further revision before publication.

**Reviewer Concerns:**

- Invariance to roto translation was addressed
- Writing quality was partially addressed
- Comparisons to more sophisticated baselines were not significantly addressed

**Reviewer Scores:**

From the discussions some significant concerns of rRq4 were addressed and it is possible this reviewer would increase their score. The other reviewers would speculatively maintain their score.

---

### Decision · Program_Chairs · 2026-01-26

Reject